# Applying Mixture of Municipal Incinerator Bottom Ash and Sewage Sludge Ash for Ceramic Tile Manufacturing

**DOI:** 10.3390/ma14143863

**Published:** 2021-07-10

**Authors:** Deng-Fong Lin, Wei-Jhu Wang, Chia-Wen Chen, Kuo-Liang Lin

**Affiliations:** Department of Civil Engineering, I-Shou University, Kaohsiung City 840203, Taiwan; dflin@isu.edu.tw (D.-F.L.); tony960222@isu.edu.tw (W.-J.W.); bear219219@gmail.com (C.-W.C.)

**Keywords:** municipal incinerator bottom ash (MIBA), sewage sludge ash (SSA), scanning electron microscope (SEM), nuclear magnetic resonance (NMR), ceramic manufacturing

## Abstract

Municipal incinerator bottom ash (MIBA) and sewage sludge ash (SSA) are secondary wastes produced from municipal incinerators. Landfills, disposal at sea, and agricultural use have been the major outlets for these secondary wastes. As global emphasis on sustainability arises, many have called for an increasing reuse of waste materials as valuable resources. In this study, MIBA and SSA were mixed with clay for ceramic tile manufacturing in this study. Raw materials firstly went through TCLP (Toxicity Characteristic Leaching Procedure) to ensure their feasibility for reuse. From scanning electron microscopy (SEM), clay’s smooth surface was contrasted with the porous surface of MIBA and SSA, which led to a higher water requirement for the mixing. Specimens with five MIBA mix percentages of 0%, 5%, 10%, 15%, and 20% (wt) and three SSA mix percentages of 0%, 10%, and 20% (wt) were made to compare how the two waste materials affected the quality of the final product and to what extent. Shrinkage tests showed that MIBA and SSA contributed oppositely to tile shrinkage, as more MIBA reduced tile shrinkage, while more SSA encouraged tile shrinkage. However, as the kiln temperature reached 1150 °C, the SiO2-rich SSA adversely reduced the shrinkage due to the glass phase that formed to expand the tile instead. Both MIBA and SSA increased water tile absorption and reduced its bending strength and wear resistance. Increasing the kiln temperature could effectively improve the water absorption, bending strength, and wear resistance of high MIBA and SSA mixes, as SEM showed a more compact structure at higher temperatures. However, when the temperature reached 1100 °C, more pores appeared and seemingly exhausted the benefit brought by the higher temperature. Complex interactions between kiln temperature and MIBA/SSA mix percentage bring unpredictable performance of tile shrinkage, bending strength, and water absorption, which makes it very challenging to create a sample meeting all the specification requirements. We conclude that a mix with up to 20% of SSA and 5% of MIBA could result in quality tiles meeting the requirements for interior or exterior flooring applications when the kiln temperature is carefully controlled.

## 1. Introduction

As the main mechanism of municipal waste treatment, incineration involving the process of combustion of wastes at high temperature (usually between 900 and 1200 °C) could reduce waste volume up to 90% [1]. Although most of the organics is destroyed during combustion, the approximate 10% residuals in the form of ashes become secondary waste and still remain a big problem. This study discusses possible solutions for two major secondary wastes, municipal incinerator bottom ash (MIBA) and sewage sludge ash (SSA). MIBA is the material discharged from the moving grate of municipal solid waste incinerators, while sewage sludge is the waste material produced during the treatment of industrial or municipal wastewater, and sewage sludge ash (SSA) is the by-product produced during the combustion of dewatered sewage sludge in an incinerator. According to Taiwan Environmental Protection Administration statistics, there are over 1,000,000 ton/year of municipal incinerator residues (including 90% bottom ash and 10% fly ash) discharged from municipal solid waste incinerators island wide [1]. That is, over 900,000 tons of MIBA are produced annually. In addition, there are more than 500,000 ton/year sewage sludge ash generated in Taiwan [2]. Landfills, disposal at sea, and agricultural use have been the major outlets for these secondary wastes [3], although some countries extensively use them as fillers for road sub-base construction [4]. As global emphasis on sustainability arises, many have called for increasing reuse of waste materials as valuable resources. Establishing MIBA and SSA as viable secondary construction materials can contribute greatly to realizing this target [5].

The concrete industry, while being eager to attend to the environmental issues related to the production process such as increasing energy efficiency and the adoption of alternative raw materials, represents one of the most relevant solutions for MIBA and SSA [6,7]. MIBA can replace the granular portion of concrete, in terms of fine and coarse aggregates [8]. A replacement level of 20 to 40% on fine aggregate weight has been proved feasible, but it results in problems of lower slump, an increase in both initial and final setting time, and a reduction in compressive strength [9,10].

Artificial aggregates produced by stabilization/solidification methods in a rotary plate granulator makes it possible to substitute upgraded MIBA for coarse aggregate [11]. A maximum replacement level of around 20% in volume for structural concrete and up to 50% for plain concrete can guarantee final mechanical and durability properties to be compatible with those of ordinary concrete, at the same compressive strength. However, concrete with MIBA have shown a higher shrinkage, creep, and chloride-diffusion coefficients [12]. Other possible disadvantages of using MIBA for concrete aggregates include the decreased durability due to the high amount of heavy metals in MIBA and enhanced aggravated rebar corrosion due to the high content of chloride found in MIBA [13].

The concrete industry has also realized different applications of SSA, from cement component to concrete element in ground form. SSA has been successfully adopted as a fine filler aggregate [14]. A maximum SSA substitution of 20% in concrete production has been attempted with positive results, while SSA brings a higher water absorption capacity and porosity to the product [15]. However, when SSA is used for replacing cement, a dramatic decrease in the final strength of the hardened composites makes it a less advantageous solution. 

In addition to the concrete industry, the ceramic industry is another notable sector that has been reusing MIBA and SSA as valuable raw materials. In 2003, Cheeseman et al. from the UK attempted an early pioneering project and delivered a successful ceramic processing of incinerator bottom ash [16]. Continuing works have been done to understand the sintering behavior and properties of MIBA to produce sintered ceramics. Holmes et al. tested the compressive and flexural strengths, water absorption, and density of masonry blocks made of concrete and MIBA. They assessed the suitability of MIBA to replace 0, 10, 20, 30, 50, 75, or 100% of natural fine aggregates in masonry blocks (100 mm high × 215 mm wide × 440 mm long) with a design strength of 7 N. The results indicated that MIBA replacement levels below 20% provide adequate compression and tensile strengths with density and absorption also within satisfactory levels [17]. Bourtsalas et al. mixed a fine fraction of MIBA (under 4 mm) with glass and transformed by milling, calcining, pressing, and sintering into high-density ceramics [18]. In Italy, Schabbach et al. demonstrated the obtainment of new ceramics based on a huge amount (60 wt %) of post-treated MIBA and 40 wt % of refractory clay [19]. The treatment started from a complex process of selection and physical/mechanical treatment (aging, sieving, and washing) of MIBA, and then, an inert material with silica-based matrix, rich in iron, calcium, and aluminum oxides was obtained. When sintered between 1190 and 1240 °C, the specimens demonstrated low water absorption and high crystallinity, leading to mechanical characteristics comparable to those of commercial ceramic products (bending strength > 40 MPa).

Alongside MIBA, SSA has also been used for making ceramic products, such as bricks, tiles, and glass ceramics. As early as 1997, Endo et al. suggested using SSA for obtaining ceramic materials [20]. In 2001, Lin and Wen reported a successful brick manufactured from incinerated sewage sludge ash and clay [21]. The appropriate percentage of SSA content for producing quality bricks was in the range of 20 to 40% by weight with a 13 to 15% optimum moisture content prepared in the molded mixture and firing at 1000 °C for 6 h. With 10% ash content, the ash–clay bricks exhibited higher compressive strength than normal clay bricks. In 2002, Anderson reported a laboratory investigation into the potential utilization of incinerated sewage sludge ash to replace 5% of brick materials and delivered a comparable product with original bricks [22]. Ongoing studies have concluded that when used as contents of ceramic products such as blocks and tiles, SSA possesses similar chemical characteristics to clay and achieves the targeted densification, strength increases, and absorption reductions under high heat treatment [23,24,25]. SSA’s fluxing properties facilitate lower firing temperatures during ceramics production, although reductions in mix plasticity can require higher forming water [5]. A study in Taiwan indicated that the preferable sintering temperature to make SSA-rich tiles is 1000 °C, and its sintering time should not exceed 1 h. Higher sintering temperature would effectively lower the water absorption ratio of the final products [26]. Amin et al. successfully produced tiles meeting ISO standards with water absorption under 10% using 7% sludge ash in the mix while fired at 1150 °C [27]. 

In addition to using SSA alone to replace clay, some studies have started to add multiple waste materials for ceramic tile production to improve the quality of the final product. Lin et al. 2007 applied nano-SiO_2_ as an additive to tiles containing SSA. They proved that the tile bending strength improved with increased nano-SiO_2_ amount [28]. Lin and Cheng 2012 mixed clay and different amounts of solar panel waste glass to manufacture eco-tiles, and they used XRD (X-ray diffraction), FTIR (Fourier transform infrared spectroscopy), and SEM (scanning electron microscopy) to investigate the characteristics of the microstructures of the specimens [29]. Kim et al. 2016 used LCD (Liquid-Crystal Display) waste glass as a flux material to replace the traditional feldspar in the manufacture of the ceramic tile specimens [30]. They found that the calcined tile body containing LCD waste glass had a dense microstructure and had positive influences on tile specimens such as water absorption and the thermal expansion coefficient. Rozenstrauha et al. (2011) spotted a dense glass–ceramics composite on SSA/glass ceramics at temperatures between 1120 and 1140 °C [31]. Esmeray and Atis (2019) used sewage sludge, oven slag, and fly ash in clay brick production; however, putting sewage sludge appears to have a negative effect on the strength of the final products, so its use should not exceed 20% [32].

In this study, we attempted to mix municipal incinerator bottom ash (MIBA) and sewage sludge ash (SSA) with clay for making ceramic tiles, and we investigate how the two materials affect the quality of the final product and to what extent. Specimens with different mixes of replacement levels of MIBA, SSA, and clay were prepared and calcined at different temperatures. By conducting a series of macroscopic tests, the research team made an effort to determine appropriate combinations of raw materials at suitable sintering temperatures so that final products with a satisfactory quality level for different applications could be delivered.

## 2. Materials

### 2.1. Raw Material Properties

This section reports the basic properties of the three raw materials applied to this study. Table 1 shows the physical properties of clay, MIBA, and SSA used in this study. Physical properties were assessed according to Chinese National Standards (CNS) as specified in the table [33]. As shown in the table, the clay has the largest specific gravity, followed by MIBA and SSA. Our MIBA was obtained from local incinerated residues fired at a temperature above 850 °C. The specific gravity of MIBA was slightly smaller than that of the clay. Our sewage sludge ash was obtained from a local incinerator, which receives its sludge from a water treatment plant located near a high-tech industrial district of Kaohsiung, Taiwan. The resulted SSA appeared to have large specific surface area leading to the smallest specific gravity and largest porosity among the three raw materials. When obtained from the local supplier, some materials were with heterogeneous dimensions so that a sieving process with No. 50 (3 mm) sieve was used to obtain a more appropriate batch for future mixing. The Standard Test Method for Particle-Size Analysis of Soils according to CNS 486 (correspondent to ASTM D422) was used to determine the particle size distribution of the three raw materials, which is shown in Figure 1. The result shows that SSA had the coarsest particle sizes ranging from 0.03 to 0.3 mm, followed by MIBA with range of 0.0303–0.3 mm and lastly clay. 

### 2.2. SEM Images for Raw Materials

SEM (scanning electron microscopy) is used to show the microstructure of the materials. SEM analysis was conducted in the Micro and Nanostructure Analysis lab at I-Shou University, and the SEM equipment used was HITACHI-S4700 (Hitachi, Tokyo, Japan). The Hitachi S-4700 is a field emission scanning electron microscope (FE-SEM), which is more powerful than a standard SEM. The S-4700, under optimal conditions, can magnify images upwards of 500,000 times and resolve features to 2 nanometers. The signals generated during SEM analysis produce a two-dimensional image and reveal its internal morphology (texture). Figure 2 shows the SEM images for clay, MIBA, and SSA. The images show that the surface of clay was smooth, while more edge angles were spotted for the crystal of MIBA. Since MIBA is incinerated at high temperature and then cooled with water, a lot of pores were detected on its coarse surface. SSA was more of a powder material with irregular shapes and lots of pores. Comparing Figure 2b with Figure 2c, we concluded that the pores in MIBA were larger than those of the SSA. However, the specific surface area of SSA was larger than that of MIBA, so SSA would thus expose particles to more air, and the porosity of SSA was larger than that of MIBA. It implied that the application of SSA for tile manufacturing may increase the water absorption of tile specimens.

### 2.3. EDS Analysis

Table 2 shows the EDS (energy-dispersive X-ray spectroscopy) results of the raw materials. EDS analysis was also conducted in the Micro and Nanostructure Analysis lab at I-Shou University, and the EDS equipped with Hitachi-S4700 was HORIBA (HORIBA, Kyoto, Japan). As seen in the table, the main chemical elements contained in clay were O, Si, and Al; when O, Ca, P, Na, and C were in MIBA and O, Si, and Al were in SSA. O occupied a large percentage of all three materials, while Si occupies 27.2% of clay and 36.9% of SSA, and Ca occupies 21.2% of MIBA. MIBA also contained 8.66% of C compounds.

Oxides commonly present in ceramic materials including SiO_2_, CaO, and Al_2_O_3_, which can also be found in MIBA and SSA [5,18,19,34,35,36,37]. However, past studies have shown that MIBA and SSA are complex and heterogeneous materials with composition varying in time and location. According to our EDS results, no Ca has been detected in our clay and SSA, and it means that CaO or calcium carbonate was not a main composition for the clay and SSA in this study. Past studies have also shown similar compositions in clay [38,39,40] and in SSA [41,42,43,44]. A small amount (8.66%) of carbon was found in MIBA. As reported in many previous studies, minerals found in MIBA include quartz (SiO_2_), calcite (CaCO_3_), gehlenite (Ca_2_Al_2_SiO_7_), and hematite (Fe_2_O_3_) [16,18,19]. It could be determined that the carbon contents belonged to CaCO_3_. 

### 2.4. TCLP Test

TCLP (Toxicity Characteristic Leaching Procedure) is a chemical analysis process to determine whether hazardous elements are present in a waste material. To recycle the MIBA and SSA as acceptable construction material, their TCLP test should yield results meeting the hazardous industrial waste standards. For this purpose, we followed the Taiwan EPA TCLP method-NIEA R201.15C, which corresponds to USEPA method 1311. Table 3 shows the TCLP test results for MIBA and SSA used in this study. The results were compared with Taiwan EPA “Standards for Defining Hazardous Industrial Waste”, as shown in Table 4 [45], and concluded to be acceptable for reuse.

The largest amount of heavy metal found in MIBA was Cu (1.57 mg/L), followed by Ba (0.658 mg/L) and Cd (<0.1 mg/L). Moreover, the largest amount of heavy metal in SSA was also Cu (4 mg/L), followed also by Ba (<0.2 mg/L). No other harmful heavy metals were detectable in MIBA and SSA, suggesting that our MIBA and SSA were suitable for recycling.

## 3. Results and Discussion

Specimens with five MIBA mix percentages of 0%, 5%, 10%, 15%, and 20% (wt) and three SSA mix percentages of 0%, 10%, and 20% (wt) were made so that a total of 15 sets of specimens were prepared. Prior to making specimens, all three raw materials went through an Atterberg test and found their plasticity limits in order to determine a proper mixing water amount. During the mixing process, proper compositions of clay, MIBA, and SSA were carefully prepared and kneaded in a shaft clay mixer, and a de-airing vacuum pug mill was used to expel internal pores. Finally, the properly kneaded mixtures were placed in a mold of 12 ∗ 6 ∗ 1 cm^3^ and compressed at a 34.32 ± 0.5 MPa pressure to produce the final specimens. Thirty tile samples were contained in each set of specimens, and they were ready for use in the following tests. 

### 3.1. Atterberg Limits

To study how various MIBA and SSA replacements affect the amount of water applied, we used the Atterberg limits test to obtain plasticity limits of the different mixes according to CNS 5088 [33]. Figure 3 shows the plastic limits at different amounts of MIBA and SSA replacements. As shown in the figure, the plastic limit increased with increasing amounts of MIBA and SSA replacements, indicating more mixing water required for a larger amount of MIBA and SSA replaced in the specimens. From the discussion above, the porosity of SSA was larger than that of the MIBA. It suggests that the effects of SSA on the plastic limit was larger than that of the MIBA.

### 3.2. Shrinkage

Shrinkage tests were done according to CNS3299-2 [33]. Figure 4 shows the shrinkages for floor tiles contained with various mixes of MIBA and SSA at different kiln temperatures. The shrinkage reduced with the greater amount of MIBA mix at kiln temperature between 1000 and 1100 °C. When the kiln temperatures were at 1000 and 1050 °C, the shrinkage also increased with greater amounts of SSA mix. However, at a kiln temperature of 1100 °C, the shrinkage decreased first and then increased with the greater amount of SSA mix. SiO_2_ was the main composition for SSA, and the quartz phase and glass phase were the main structures for SiO_2_. When the glass phase occupied more in SiO_2_, the surface of the tile specimens became glassy, and the air would be trapped inside the glassy surface, producing more pores in the tile specimens during the calcined process. Hence, the tile body became expanded easily. In this study, the SSA may contain more glass phase in SiO_2_. When the kiln temperature reached 1150 °C, the tile body started to melt and a glass foam in the body was expanded, which is caused by part of the glass phase in SiO_2_, as shown in Figure 4d. Thus, the glass foam structure agreed with the findings in previous studies [46,47,48,49].

### 3.3. Weight Loss on Ignition

Figure 5 shows the weight loss on ignition of tiles with various mixes of MIBA and SSA at different kiln temperatures. It appears that the weight loss on ignition of tiles remained relatively steady and did not change a lot with increasing kiln temperature. With the increasing amount of MIBA in the mix, weight loss on ignition seemed to be increasing. The reason could be that the high amount of organic, non-organic matters, and heavy metals in MIBA could be easily burned out at higher kiln temperature. On the other hand, with increasing SSA in the mix, weight loss on ignition decreased. The large amount of SiO_2_ in SSA seemed to be building a stronger structure, which improves the melting temperature of tile specimens and becomes harder to burn out. 

### 3.4. Specific Gravity

Specific gravity was measured according to CNS 3299-3 [33]. Figure 6 shows the specific gravity of tiles with various mixes of MIBA and SSA at different kiln temperatures. The specific gravity increased as the kiln temperature rose and reached 1100 °C, but the specific gravity significantly dropped at 1150 °C. With MIBA and SSA in the mix, tile specific gravity even dropped under 1 and become floatable on the water (as shown in the Figure A1 in Appendix A). Observing the trends in Figure 6, we found that increasing the mix of MIBA decreased the specific gravity of the tile specimens. However, SSA created a complex scene, since its effect on specific gravity could be positive and negative with different mixes and different temperatures. No regular pattern could be determined for SSA effects on specific gravity. 

### 3.5. Water Absorption

Water absorption tests were conducted according to CNS3299-3 [33]. Figure 7 shows results of the water absorption tests for tiles with various mixes of MIBA and SSA at different kiln temperatures. At the same kiln temperature, water absorption increased with increasing amounts of MIBA and SSA mixes. In addition, it appears that water absorption reduced with the increasing of kiln temperature, especially when fired over the temperature of 1100 °C. When the kiln temperature reached 1150 °C, tile surface became vitreous and shiny, and it prevented water penetration.

### 3.6. Bending Strength

Bending strength tests were conducted according to CNS3299-4 [33]. Figure 8 shows the bending strength tests for tiles with various mixes of MIBA and SSA at different kiln temperatures. Since both MIBA and SSA were categorized as porous materials, the porosity of the tiles must increase with MIBA and SSA replacements. The increasing porosity weakened the interior structure of tile specimens and resulted in a lowered bending strength. On the other hand, higher kiln temperatures could produce a denser interior structure of floor tiles. Bending strength increased with increasing kiln temperature within the range of 1000–1100 °C and achieved the maximum strength at 1100 °C. However, when the kiln temperature reached 1150 °C, foam was produced in the tile specimens, and the glass phase in the SiO_2_ of SSA produced large pores by air trapped in the tile specimens, resulting in a dramatic bending strength reduction. 

### 3.7. Wear Resistance

Wear resistance tests were conducted according to ASTM D968. Figure 9 shows the wear resistance tests for tiles with various mixes of MIBA and SSA at different kiln temperatures. As shown in the figure, since MIBA is a porous material, adding MIBA could harm the compactness of floor tiles. The amount of wear increased with the increasing amount of MIBA, as expected. SSA’s effects on tile wear resistance seemed much more complicated. When fired at 1000 and 1050 °C, it appears that SiO_2_ in SSA improved the hardness of floor tile specimens, and the amount of wear decreased with increasing amounts of SSA. At 1100 °C, the amount of wear increased first and then reduced with the increasing amount of SSA replacement. When the kiln temperature reached 1150 °C, melting of the tile specimens was observed, and foam was produced in the tile specimens. The amount of tile wear increased at this kiln temperature. Furthermore, the glass phase produced by SiO_2_ in SSA could trap air inside tile specimens and affect the compactness of the tile body. The amount of tile wear increased with the increasing amount of SSA at 1150 °C.

### 3.8. SEM Analysis

Figure 10, Figure 11, Figure 12 and Figure 13 show SEM images for floor tile specimens contained with 10 and 20% SSA replacements and 0 to 20% MIBA replacements firing at various kiln temperatures. 

From Figure 10 with only SSA replacement at 10%, it can be found that tiles are getting more compact as the kiln temperature rises, but it seems that the pores have increased as the kiln temperature reaches 1150 °C. Figure 11 shows the same trends with SSA replacement at 20%, and again, the pores seem to keep getting bigger at 1150 °C, while the compactness is increasing with higher kiln temperature. Figure 12 shows the SEM scenes when MIBA is adding to the mix. It is obvious that as the MIBA is increased, the compactness of the tiles is decreased. Figure 13 shows the same mix of 10% SSA and 20% MIBA at different kiln temperatures. The microstructure of the tiles appears to be more compact, but pores seem to be growing at 1150 °C.

### 3.9. EDS Analysis

Figure 14 and Figure 15 show EDS images for floor tiles with 10 and 20% SSA replacements and 0 to 20% MIBA replacements at kiln temperature of 1100 °C. The amount of Si decreased and increased with the increasing amount of MIBA and SSA replacements, respectively. The kiln temperature had no apparent influence on the amount change of Si. However, when the temperature reached 1100 °C, the amount of Si reduced with increasing amounts of SSA replacement because the glass phase structure produced from SiO_2_ was formed, and a shiny surface of tile specimens was observed. 

### 3.10. XRD Analysis

Table 4 shows XRD (X-ray diffraction) analysis results and Figure 16 shows the f XRD diffractograms or tiles with various mixes of MIBA and SSA at different kiln temperatures. It shows that the dominant component of tiles was SiO_2_. Since the dominant component of SSA was also SiO_2_, the amount of SiO_2_ in tiles increased with the increasing amount of SSA added. Compared with results of bending strength tests, when the amount of Ca(Al_2_Si_2_O_8_) produced from the combination of SiO_2_, CaO, and Al_2_O_3_ increased, the bending strength of tile specimens reduced. When same kiln temperature was considered, it could come to the conclusion that the bending strength decreased with more MIBA added. 

### 3.11. Si-NMR Analysis

Si-NMR (^29^Si Nuclear Magnetic Resonance) is used to examine structural properties of porous silicon compounds in the tile specimens. The superscript n of Q^n^ refers to the number of connections of a silicon atom to other silicon species. The Q^0^ type refers to a silicon atom with no connections to another silicon species, the Q^1^ type refers to a silicon atom connected to one silicon O-Si species, and so forth. Figure 17 shows the integration results of Si-NMR spectra analysis for floor tiles that contain 0–20% SSA and 0% MIBA replacements firing at kiln temperature of 1050 °C. It appears that there was a peak observed at −108 ppm for Q^4^. The peak value of Q^4^ reduced with more SSA. The values of Q^4^ after integration increased with more SSA added. It suggests that the addition of SSA helped improve the development of SiO_4_ tetrahedral structure. 

The results obtained from Si-NMR analysis for floor tiles that contained 20% SSA and 20% MIBA replacements at various kiln temperatures also show that the peak values for Q^4^ became apparent with increasing kiln temperature. It suggests that the increase of kiln temperature helped Si atoms improve the development of silicate structures. Figure 18 shows the integration results of Si-NMR spectra analysis for floor tiles that contained 20% SSA and 20% MIBA firing at various kiln temperatures. The values of Q^4^ after integration increased first and then reduced with increasing kiln temperature. The highest value of Q^4^ was obtained at 1100 °C. It suggests that the kiln temperature of 1100 °C helped improve the development of SiO_4_ tetrahedral structure. This result conformed with those obtained from bending strength tests.

### 3.12. Quality Classification

According to CNS 9737 R1018 [50], bending strength, shrinkage, and water absorption percentage are the criteria to determine if the tile meet the qualification rating. The minimum bending strength for interior tiles and exterior tiles is 540 N and 1080 N, respectively. Shrinkage for interior tiles is limited to ±1.0 mm for length, ±0.8 mm for width, and ±0.5 mm for thickness, while shrinkage for exterior tiles is limited to ±2.0 mm for length, ±1.6 mm for width, and ±1.2 mm for thickness. The water absorption rating is categorized as four types: Ia (<0.5%), Ib (<3.0%), II (<10.0%), and III (50%). Table 5 summarizes the quality classification for ceramic floor tiles with various MIBA and SSA mixes. With increasing amounts of MIBA, it became more difficult to meet the bending strength requirement for interior tiles. Increasing MIBA resulted in an increase of porosity in tile structure and a significant reduction on qualified rating for tile specimens. At a kiln temperature of 1100 °C, the bending strength requirement for both the interior and exterior floor can be met with different amounts of SSA and MIBA mixes, but unfortunately, tile shrinkage becomes too large to allow sufficient control of tile size for possible application. When the kiln temperature reached 1150 °C, all but one tile specimens were not strong enough and had a hard time meeting the required bending strength in the specifications. The possible cause was the SSA contained with SiO_2_, which resulted in a production of glass phase on the surface of specimens that trapped air and produced pores at the interior of the tile body. In this study, the water absorption for all floor tile specimens met the standard of type III, and the qualifying rate increased with increasing kiln temperature. However, the qualifying rate reduced with increasing amounts of MIBA and SSA mixes. Table 5 summarizes the quality classification for ceramic floor tiles with various MIBA and SSA mixes.

Table 6 shows the favorable mix designs meeting the requirements in the specifications. As shown in the table, tiles containing 10% SSA and 0% MIBA fired at 1150 °C met the water absorption requirements set for the interior ceramic floor tile standards of type Ib. Moreover, tiles with 20% SSA and 0% MIBA fired at 1050 °C met the requirements set for both interior and exterior ceramic floor tile standards with acceptable water absorption ratings of II and III. Tile specimens with 0–5% MIBA fired at 1100 °C met the requirements set for the interior ceramic floor tile standards with a water absorption rating at type II.

## 4. Conclusions

As SSA and MIBA have been proven to be feasible replacement materials for ceramic production, the study makes a pioneering attempt to mix both materials. The interactions between SSA and MISA at various kiln temperature are complicated, but this study has come up with some encouraging results, and numerous feasible applications are suggested in Table 5. It concludes that a mix with up to 20% of SSA and 5% of MIBA could result in quality tiles complying with specifications for interior or exterior flooring applications at certain kiln temperatures. The idea of reusing waste materials has a double environmental and economic benefit: the avoidance of additional waste disposal expenses and reduction of ceramic raw materials use. No cost is required for obtaining MIBA and SSA, although some pre-treatment and testing costs for waste materials may occur due to local regulations. However, these additional costs can easily be compensated by the saving from avoidance of waste disposal. This study concludes that a 25% raw material saving could be achieved with 20% of SSA and 5% of MIBA replacement from the economic point of view. Since our raw materials have passed TCLP (Toxicity Characteristic Leaching Procedure) to ensure their feasibility for reuse, we hypothesized that the tiles made with 20% SSA and 5% MIBA should not exceed TCLP limits. This study also draws the following conclusions: The application of MIBA could reduce the shrinkage rate of floor tile specimens. On the other hand, due to the fact that the SSA contained SiO_2_, the use of SSA could increase the shrinkage rate of floor tile specimens within a kiln temperature of 1000–1050 °C. However, when kiln temperature reached 1150 °C, the glass phase surface formed, the tile bodies were expanded, and the shrinkage rate was reversed.Since a large amount of organic, non-organic matters, and heavy metals in MIBA were easily burned at high kiln temperatures, the weight loss on ignition of floor tile specimens increased with more MIBA. In contrast, a large amount of SiO_2_ in SSA could improve the melting temperature of tile specimens and was hard to be burnt out; the weight loss on ignition of tile specimens increased with more SSA.In general, the water absorption of tile specimens increased with increasing amounts of MIBA and SSA, but the water absorption reduced with the increasing kiln temperature. When the kiln temperature reached 1150 °C, the tile surface became vitreous and shiny, and it prevented water penetration, so that the water absorption was reduced to close zero.The bending strength of floor tile specimens reduced with more MIBA and SSA replacements. Moreover, the tile bending strength increased with increasing kiln temperature within the range of 1000–1100 °C and would achieve the maximum strength at 1100 °C. However, when the kiln temperature reached 1150 °C, foam was produced in the tile specimens and so forth, resulting in the reduction of bending strength.Tile wear increased with more MIBA replacement, while the tile wear decreased with more SSA replacement at kiln temperatures of 1000 and 1050 °C. The amount of wear increased first and then reduced with more SSA replacement at 1100 °C. However, when the kiln temperature reached 1150 °C, the amount of tile wear increased. The amount of tile wear for tile specimens increased with more SSA and MIBA replacements at 1150 °C.When the kiln temperature was considered as the main parameter, in general, the compactness of the tile body was improved by the increasing kiln temperature. At a kiln temperature of 1100 °C, the amount of pores increased with more MIBA and SSA replacements, and more pores were observed in the tile bodies from SEM images.The results obtained from Si-NMR analysis show that the peak values for Q_4_ became apparent with increasing kiln temperature. It suggests that the increase of kiln temperature helped Si atoms improve the development of the silicate structure. The values of Q_4_ after integration increased first and then reduced with increasing kiln temperature. The highest value of Q_4_ was obtained at 1100 °C. It suggests that the kiln temperature of 1100 °C helped improve the development of a SiO_4_ tetrahedral structure. This result conformed with that obtained from bending strength tests.

## Figures and Tables

**Figure 1 materials-14-03863-f001:**
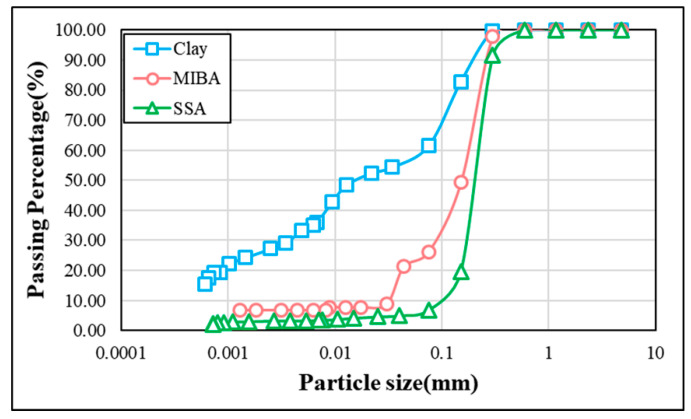
Particle distribution of the raw materials.

**Figure 2 materials-14-03863-f002:**
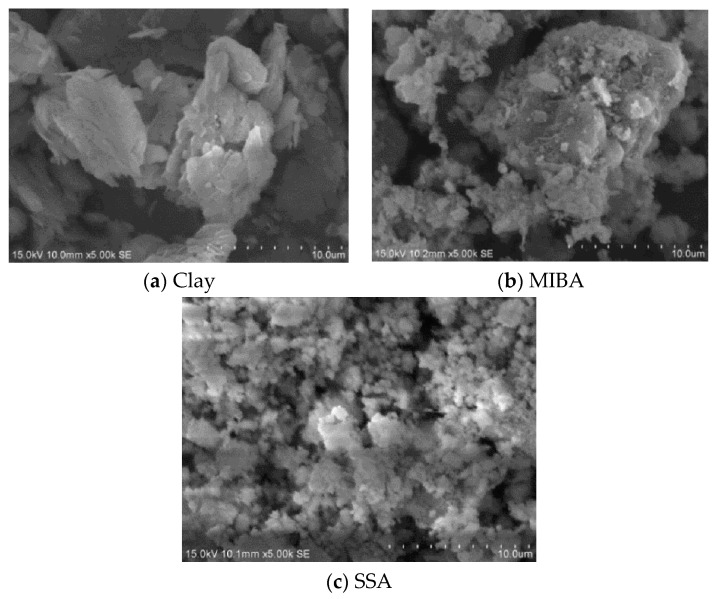
SEM images for the raw materials. (**a**)Clay; (**b**) MIBA; (**c**) SSA.

**Figure 3 materials-14-03863-f003:**
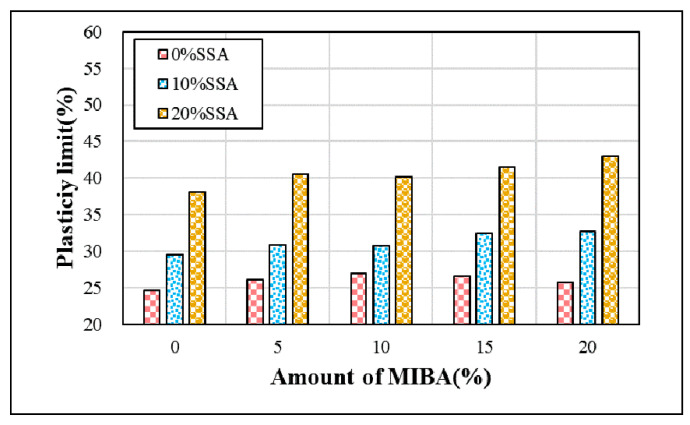
Plasticity limits of the specimens.

**Figure 4 materials-14-03863-f004:**
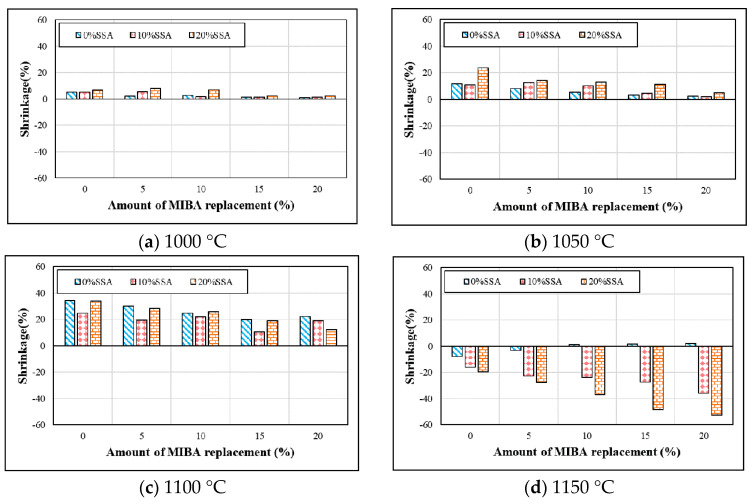
Shrinkage at different sintering temperature. (**a**) 1000 °C; (**b**) 1050 °C; (**c**) 1100 °C; (**d**) 1150 °C.

**Figure 5 materials-14-03863-f005:**
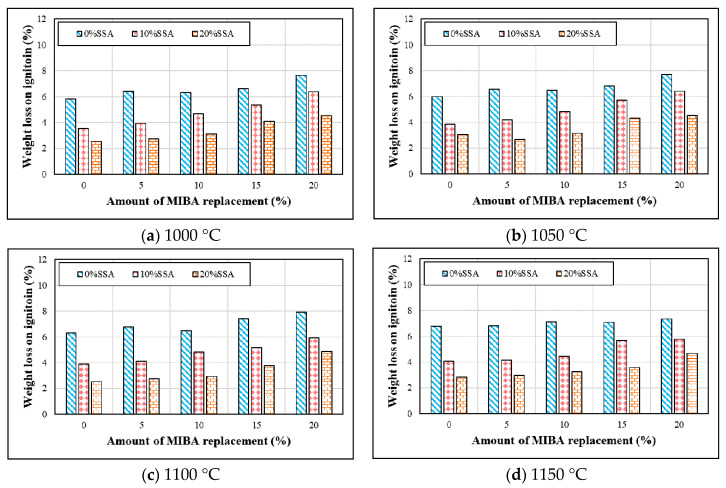
Weight loss ignition at different sintering temperature. (**a**) 1000 °C; (**b**) 1050 °C; (**c**) 1100 °C; (**d**) 1150 °C.

**Figure 6 materials-14-03863-f006:**
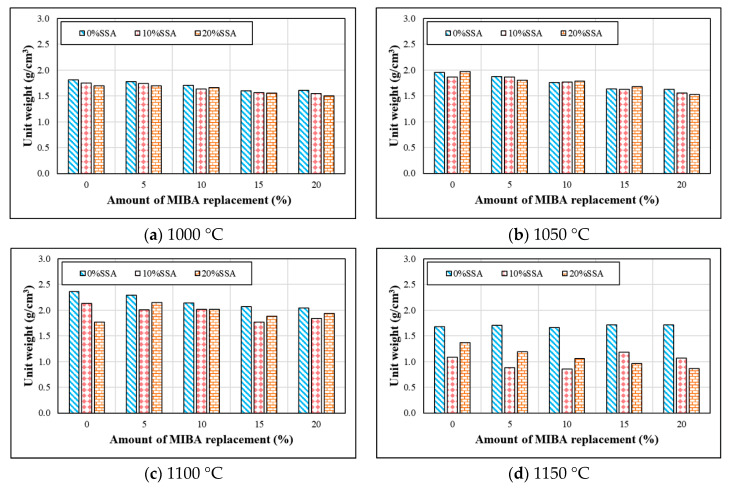
Specific gravity of specimens at different sintering temperature. (**a**) 1000 °C; (**b**) 1050 °C; (**c**) 1100 °C; (**d**) 1150 °C.

**Figure 7 materials-14-03863-f007:**
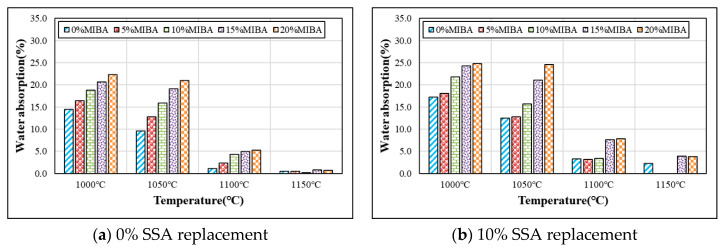
Results of the water absorption tests. (**a**) 0% SSA; (**b**) 10% SSA; (**c**) 20% SSA.

**Figure 8 materials-14-03863-f008:**
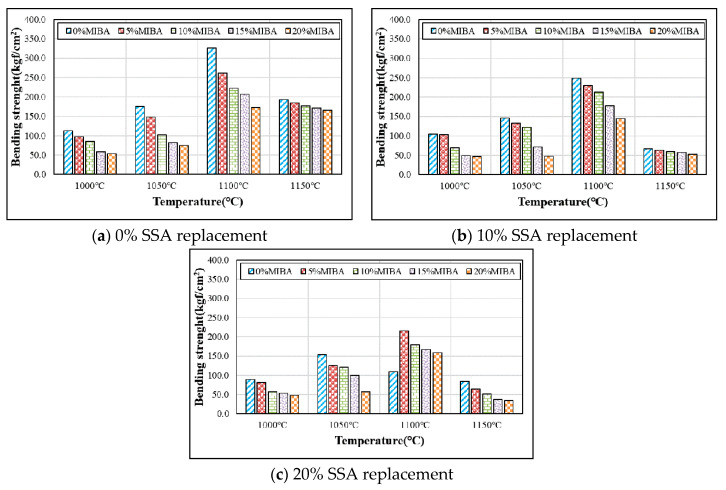
Results of the bending strength tests. (**a**) 0% SSA; (**b**) 10% SSA; (**c**) 20% SSA.

**Figure 9 materials-14-03863-f009:**
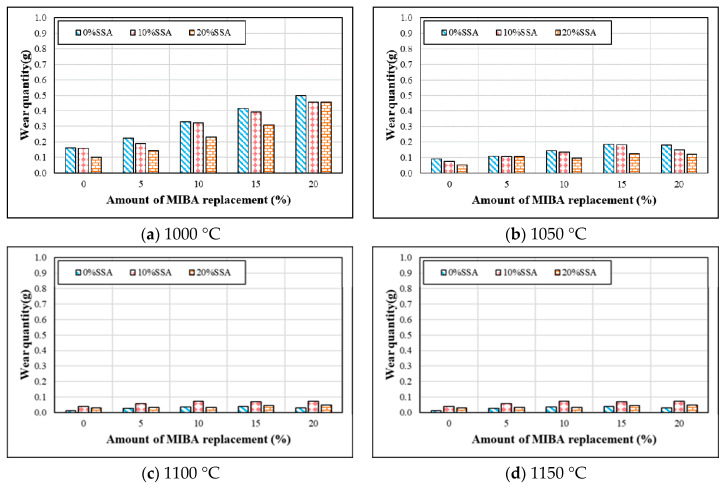
Wear resistance of specimens at different sintering temperature. (**a**) 1000 °C; (**b**) 1050 °C; (**c**) 1100 °C; (**d**) 1150 °C.

**Figure 10 materials-14-03863-f010:**
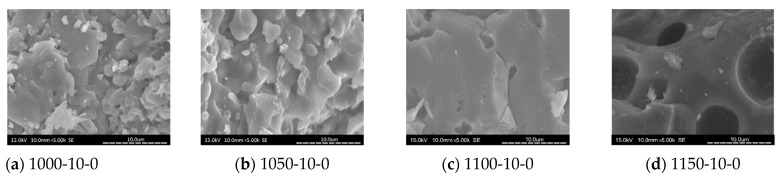
SEM of 10% SSA replacement and no MIBA at different kiln temperatures. (**a**) 1000 °C-10% SSA-0% MIBA; (**b**) 1050 °C-10% SSA-0% MIBA; (**c**) 1100 °C-10% SSA-0% MIBA; (**d**) 1150 °C-10% SSA-0% MIBA.

**Figure 11 materials-14-03863-f011:**
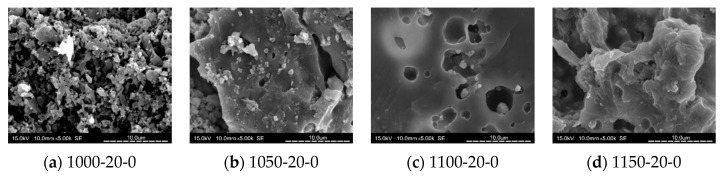
SEM of 20% SSA replacement and no MIBA at different kiln temperatures. (**a**) 1000 °C-20% SSA-0% MIBA; (**b**) 1050 °C-20% SSA-0% MIBA; (**c**) 1100 °C-20% SSA-0% MIBA; (**d**) 1150 °C-20% SSA-0% MIBA.

**Figure 12 materials-14-03863-f012:**
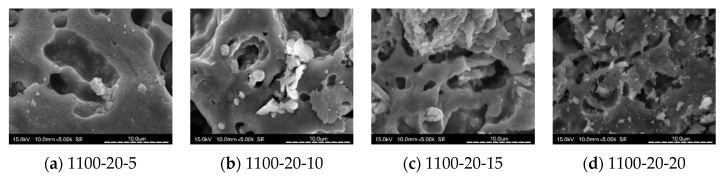
SEM of 20% SSA replacement and various MIBA at 1100 °C. (**a**) 1100 °C-20% SSA-5% MIBA; (**b**) 1100 °C-20% SSA-10% MIBA; (**c**) 1100 °C-20% SSA-15% MIBA; (**d**) 1100 °C-20% SSA-20% MIBA.

**Figure 13 materials-14-03863-f013:**
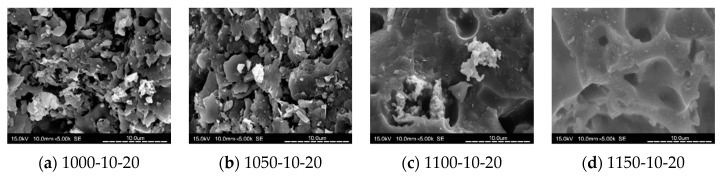
SEM of 10% SSA and 20% MIBA replacement at different kiln temperatures. (**a**) 1000 °C-10% SSA-20% MIBA; (**b**) 1050 °C-10% SSA-20% MIBA; (**c**) 1100 °C-10% SSA-20% MIBA; (**d**) 1150 °C-10% SSA-20% MIBA.

**Figure 14 materials-14-03863-f014:**
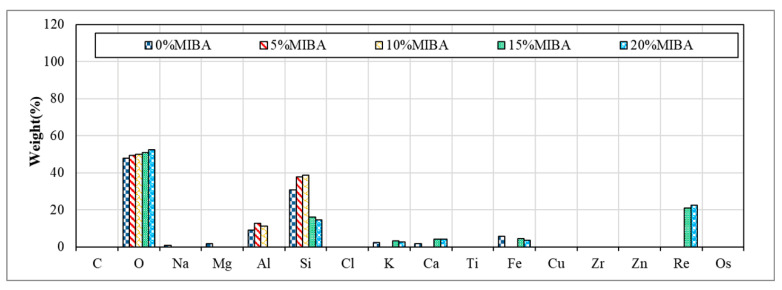
EDS of 10% SSA and various MIBA replacements at 1100 °C kiln temperature.

**Figure 15 materials-14-03863-f015:**
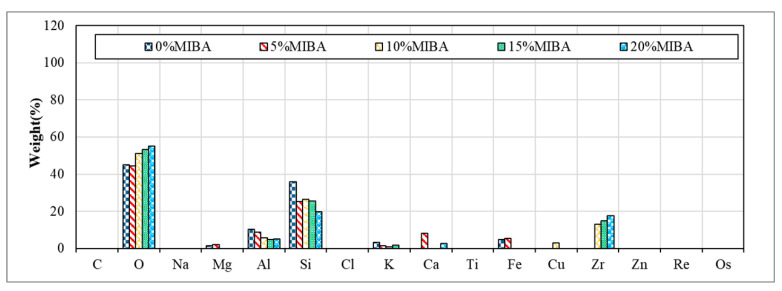
EDS of 20% SSA and various MIBA replacements at 1100 °C kiln temperature.

**Figure 16 materials-14-03863-f016:**
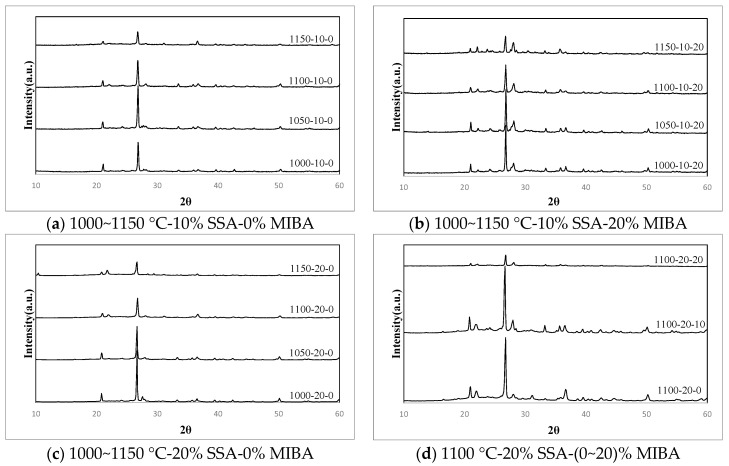
XRD diffractograms. (**a**) 1000~1150 °C-10% SSA-0% MIBA; (**b**) 1000~1150 °C-10% SSA-20% MIBA; (**c**) 1000~1150 °C-20% SSA-0% MIBA; (**d**) 1100 °C-20% SSA-(0~20) %MIBA.

**Figure 17 materials-14-03863-f017:**
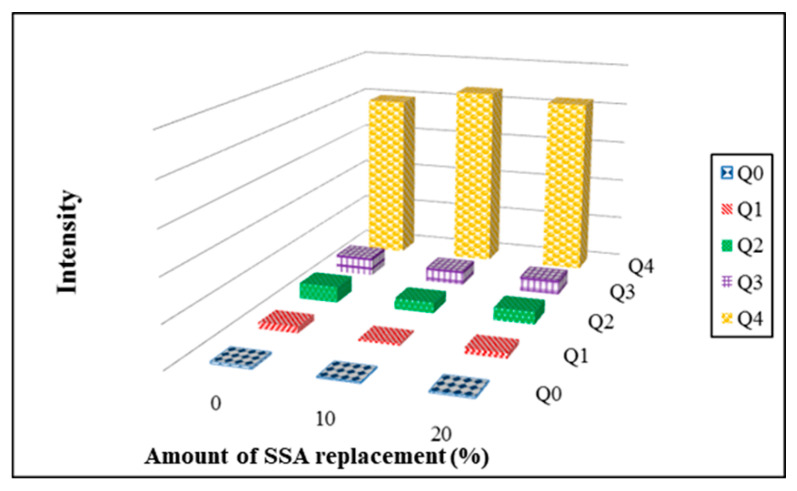
Si-NMR results of various SSA replacement at 1050 °C kiln temperature.

**Figure 18 materials-14-03863-f018:**
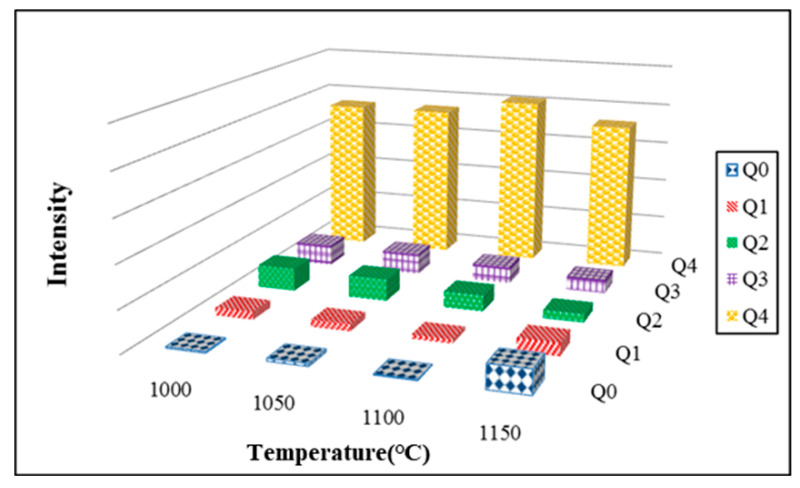
Si-NMR results of 20% SSA and 20% MIBA at various kiln temperatures.

**Table 1 materials-14-03863-t001:** Physical properties of the three raw materials.

	Clay	MIBA	SSA	Standard *
Specific gravity	2.65	2.41	2.00	CNS 5090
Unit weight (kg/m^3^)	1251.83	1004.62	698.45	CNS 1183
Pore ratio (%)	52.84	58.33	65.14	CNS 1163
Specific surface area (cm^2^/g)	2562.48	1276.41	3441.27	CNS 2924

* All CNS methods referred can be found in [33].

**Table 2 materials-14-03863-t002:** EDS results of the raw materials.

Element	C	O	Na	Al	Si	P	S	Cl	K	Ca	Fe	Br	Zr	Cu
Clay%	-	49.5	-	10.2	27.2	-	-	-	3.43	-	4.25	-	5.47	-
MIBA%	8.66	43.5	8.67	-	1.34	11.6	1.77	1.45	-	21.2	-	1.88	-	-
SSA%	-	43	-	9.28	36.9	-	-	-	-	-	-	-	6.79	3.41

**Table 3 materials-14-03863-t003:** TCLP test results for MIBA and SSA.

Element	Cu	Ba	Cd	As	Pb	Cr	Hg	Cr^6+^	Se
SSA(mg/L)	4	<0.2	ND	ND	ND	ND	ND	ND	ND
MIBA(mg/L)	1.57	0.658	<0.100	ND	ND	ND	ND	ND	ND
Standard	≤12.0	≤10.0	≤4.0	≤0.4	≤4.0	≤0.8	≤0.016	≤0.2	≤0.8

ND: not detected.

**Table 4 materials-14-03863-t004:** XRD of 10%, 20% SSA, and 0% to 20% MIBA at various kiln temperatures.

SSA%	Temperature	MIBA %	SiO_2_	AlPO_4_	CaSiO_3_	MgSiO_3_	Ca(Al_2_Si_2_O_8_)	K(AlSi_3_O_8_)
10%	1000 °C	0%	47.1	1.3	5.6	18.0	16.8	11.3
		10%	51.9	1.7	5.7	17.7	16.1	6.8
		20%	40.7	0.5	6.3	27.6	19.6	5.2
	1050 °C	0%	56.3	0.3	8.1	8.7	21.3	5.2
		10%	53.4	1.2	-	26.1	19.3	-
		20%	42.9	1.0	-	16.7	33.7	5.8
	1100 °C	0%	97.2	2.8	-	-	-	-
		10%	62.0	-	-	-	38.0	-
		20%	51.3	-	-	-	48.7	-
	1150 °C	0%	54.5	2.4	-	22.1	11.5	9.5
		10%	45.1	0.4	-	30.9	15.9	7.7
		20%	33.9	16.5	-	-	49.6	-
20%	1000 °C	0%	88.3	0.3	11.0	0.4	-	-
		10%	48.8	1.0	13.3	18.2	18.8	-
		20%	57.0	1.2	-	-	41.7	-
	1050 °C	0%	87.3	0.9	8.6	3.2	-	-
		10%	56.6	0.9	11.6	11.2	19.7	-
		20%	52.5	0.0	-	15.6	28.0	3.9
	1100 °C	0%	66.9	1.2	-	19.2	12.8	-
		10%	48.8	1.4	-	28.2	21.7	-
		20%	32.0	11.0	6.2	26.2	24.7	-
	1150 °C	0%	55.8	1.0	-	24.7	18.5	-
		10%	33.4	-	-	33.4	33.2	-
		20%	24.1	0.5	10.6	38.6	26.2	-

**Table 5 materials-14-03863-t005:** Quality compliance summary.

Material			Temperature (°C)	Judgment Criteria							
				Interior Floor Tile		Exterior Floor Tile		Water Absorption (%)			
Clay (%)	SSA (%)	MIBA (%)		Bending Strength	Size Shrinkage	Bending Strength	Size Shrinkage	Ia	Ib	II	III
100	0	0	1000	**○**	**×**	**×**	**×**	**×**	**×**	**×**	**○**
			1050	**○**	**×**	**○**	**○**	**×**	**×**	**○**	**○**
			1100	**○**	**×**	**○**	**×**	**×**	**○**	**○**	**○**
			1150	**○**	**×**	**○**	**○**	**○**	**○**	**○**	**○**
95	0	5	1000	**○**	**×**	**×**	**×**	**×**	**×**	**×**	**○**
			1050	**○**	**×**	**○**	**○**	**×**	**×**	**×**	**○**
			1100	**○**	**○**	**○**	**○**	**×**	**○**	**○**	**○**
			1150	**○**	**×**	**○**	**○**	**×**	**○**	**○**	**○**
90	0	10	1000	**○**	**×**	**×**	**○**	**×**	**×**	**×**	**○**
			1050	**○**	**×**	**×**	**×**	**×**	**×**	**×**	**○**
			1100	**○**	**×**	**○**	**×**	**×**	**×**	**○**	**○**
			1150	**○**	**×**	**○**	**○**	**○**	**○**	**○**	**○**
85	0	15	1000	**×**	**×**	**×**	**○**	**×**	**×**	**×**	**○**
			1050	**○**	**○**	**×**	**○**	**×**	**×**	**×**	**○**
			1100	**○**	**×**	**○**	**×**	**×**	**×**	**○**	**○**
			1150	**○**	**×**	**○**	**○**	**×**	**○**	**○**	**○**
90	10	0	1000	**○**	**×**	**×**	**○**	**×**	**×**	**×**	**○**
			1050	**○**	**×**	**○**	**×**	**×**	**×**	**×**	**○**
			1100	**○**	**×**	**○**	**×**	**×**	**×**	**○**	**○**
			1150	**○**	**○**	**○**	**×**	**×**	**○**	**○**	**○**
85	10	5	1000	**○**	**×**	**×**	**○**	**×**	**×**	**×**	**○**
			1050	**○**	**×**	**×**	**○**	**×**	**×**	**×**	**○**
			1100	**○**	**×**	**○**	**×**	**×**	**×**	**○**	**○**
			1150	**○**	**×**	**×**	**○**	**×**			
80	10	10	1000	**×**	**×**	**×**	**○**	**×**	**×**	**×**	**○**
			1050	**○**	**×**	**×**	**○**	**×**	**×**	**×**	**○**
			1100	**○**	**×**	**○**	**×**	**×**	**×**	**○**	**○**
			1150	**○**	**×**	**×**	**○**	**×**			
75	10	15	1000	**×**	**×**	**×**	**○**	**×**	**×**	**×**	**○**
			1050	**○**	**×**	**×**	**○**	**×**	**×**	**×**	**○**
			1100	**○**	**×**	**○**	**×**	**×**	**×**	**○**	**○**
			1150	**○**	**×**	**×**	**○**	**×**	**×**	**○**	**○**
70	10	20	1000	**×**	**×**	**×**	**○**	**×**	**×**	**×**	**○**
			1050	**×**	**×**	**×**	**○**	**×**	**×**	**×**	**○**
			1100	**○**	**×**	**○**	**×**	**×**	**×**	**○**	**○**
			1150	**○**	**×**	**×**	**○**	**×**	**×**	**○**	**○**
80	20	0	1000	**○**	**×**	**×**	**○**	**×**	**×**	**×**	**○**
			1050	**○**	**○**	**○**	**○**	**×**	**×**	**○**	**○**
			1100	**○**	**×**	**○**	**×**	**×**	**○**	**○**	**○**
			1150	**○**	**×**	**×**	**○**	**×**	**○**	**○**	**○**
75	20	5	1000	**○**	**×**	**×**	**○**	**×**	**×**	**×**	**○**
			1050	**○**	**×**	**×**	**○**	**×**	**×**	**×**	**○**
			1100	**○**	**○**	**○**	**×**	**×**	**×**	**○**	**○**
			1150	**○**	**×**	**×**	**○**	**×**	**○**	**○**	**○**
70	20	10	1000	**×**	**×**	**×**	**○**	**×**	**×**	**×**	**○**
			1050	**○**	**×**	**×**	**○**	**×**	**×**	**×**	**○**
			1100	**○**	**×**	**○**	**×**	**×**	**×**	**○**	**○**
			1150	**○**	**×**	**×**	**○**	**×**	**○**	**○**	**○**
65	20	15	1000	**×**	**×**	**×**	**○**	**×**	**×**	**×**	**○**
			1050	**○**	**×**	**×**	**○**	**×**	**×**	**×**	**○**
			1100	**○**	**×**	**○**	**×**	**×**	**×**	**○**	**○**
			1150	**○**	**×**	**×**	**○**	**×**			
60	20	20	1000	**×**	**×**	**×**	**○**	**×**	**×**	**×**	**○**
			1050	**×**	**×**	**×**	**○**	**×**	**×**	**×**	**○**
			1100	**○**	**×**	**○**	**×**	**×**	**×**	**○**	**○**
			1150	**○**	**×**	**×**	**○**	**×**			

**○**: Sample meeting quality requirement; **×**: Sample failing quality requirement.

**Table 6 materials-14-03863-t006:** Suggested application chart.

Material			Temperature (°C)	Applications (Interior Floor Tile/Exterior Floor Tile)	Water Absorption (Ia, Ib, II, III)
Clay (%)	SSA (%)	MIBA (%)			
100	0	0	1050	Exterior floor tile	II, III
			1150	Exterior floor tile	Ia, Ib, II, III
95	0	5	1050	Exterior floor tile	III
			1100	Interior floor tile/Exterior floor tile	Ib, II, III
			1150	Exterior floor tile	Ib, II, III
90	0	10	1150	Exterior floor tile	Ia, Ib, II, III
85	0	15	1050	Interior floor tile	III
			1150	Exterior floor tile	Ib, II, III
90	10	0	1150	Interior floor tile	Ib, II, III
80	20	0	1050	Interior floor tile/Exterior floor tile	II, III
75	20	5	1100	Interior floor tile	II, III

## Data Availability

All data included in this study are available upon request by contact with the corresponding author.

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
