# Peer review of "Applying Mixture of Municipal Incinerator Bottom Ash and Sewage Sludge Ash for Ceramic Tile Manufacturing"

_materials, 2021, doi:10.3390/ma14143863_

Round 1

Reviewer 1 Report

The description of the analytical technique (sieving, aerometry, laser diffraction, etc.) to determine the particle size curve and any reference to sample preparation is missing.

Furthermore, MIBA are materials with very heterogeneous dimensions usually with the fraction <2mm around 35% of the total. Does the work take into consideration only the fine fraction of this material? In this case how was it separated?

It would be useful to have the granulometric curves of the original MIBA and SSA materials, with the percentages of the different passers-by and the upper and lower dimensional limits and the fractions actually used in the research.

The analyzes in EDS show a C content of more than 8% in MIBA. It would be advisable to establish the compounds of C through other analytical techniques (XRD and FTIR-TGA) in order to establish in which mineralogical (inorganic) phases or in which (organic) compounds they are formed with carbon. In the subsequent stages of heat treatment to obtain the final products, no degassing phenomena occur? This part is fundamental for a research that involves the reuse of waste materials.

There is a lack of essential information for TLCP tests. In particular, the method of execution (amount of material used, L / S ratio, leaching time) and the analytical techniques used to analyze the eluate. Finally, the legislative references are missing.

The technical characterization of the materials obtained is good but the part relating to the possible toxicity of the products made is missing.

The work is of interest as it explores the reuse of waste materials from a circular economy perspective. To obtain this result, however, it is important to determine the end of waste condition, satisfying the conditions of toxicity and physical-mechanical requirements of the raw materials to be replaced.

It is therefore essential to carry out the complete characterization of the materials before their reuse (MIBA, SSA) and that of the final product; in the case of this work, it is therefore necessary to better characterize the MIBA and SSA products and the finished product by subjecting it to release tests.

In particular: 

Line 124: what is the standard for industrial waste used in the table? Please add reference to the legislation. The standard concentration are expressed in mg/L?

Line 140: cm3, correct using superscript.

Figure 4, 5, 6, 8, 9, 10: improve the resolution and the quality of the figure.

Paragraph 4, conclusions: correct oC into °C.

Author Response

Reviewer 1

The description of the analytical technique (sieving, aerometry, laser diffraction, etc.) to determine the particle size curve and any reference to sample preparation is missing.

Standard Test Method for Particle-Size Analysis of Soils according to CNS 486 (complying with ASTM D422) were used to determine the particle size distribution of the three raw materials as described in Section 2.1.  Other methods used for assessing physical properties were also added in Table 1.

üFurthermore, MIBA are materials with very heterogeneous dimensions usually with the fraction <2mm around 35% of the total. Does the work take into consideration only the fine fraction of this material? In this case how was it separated?

When obtained from the local supplier, some materials, especially MIBA, appeared to be with heterogeneous dimensions, so a sieving process with No. 50 (3mm) sieve was used to obtain a more appropriate batch for future mixing. (also added in section 2.1)

üIt would be useful to have the granulometric curves of the original MIBA and SSA materials, with the percentages of the different passers-by and the upper and lower dimensional limits and the fractions actually used in the research.

Only materials passing No.50 sieve were used in the study, as shown in Figure 1.

 üThe analyzes in EDS show a C content of more than 8% in MIBA. It would be advisable to establish the compounds of C through other analytical techniques (XRD and FTIR-TGA) in order to establish in which mineralogical (inorganic) phases or in which (organic) compounds they are formed with carbon. In the subsequent stages of heat treatment to obtain the final products, no degassing phenomena occur? This part is fundamental for a research that involves the reuse of waste materials.

As reported in previous studies, minerals found in MIBA include quartz (SiO2), calcite (CaCO3), gehlenite (Ca2Al2SiO7) and hematite (Fe2O3).  It could be determined that the Carbon contents belong to CaCO3. Citation is added to support the argument.

üThere is a lack of essential information for TLCP tests. In particular, the method of execution (amount of material used, L / S ratio, leaching time) and the analytical techniques used to analyze the eluate. Finally, the legislative references are missing.

We follow Taiwan EPA NIEA R201.15C for TCLP, which is correspondent to USEPA method 1311.  (added in Section 2.4).  The results were compared with Taiwan EPA “Standards for Defining Hazardous Industrial Waste” table 4 and found to be acceptable.

üThe technical characterization of the materials obtained is good but the part relating to the possible toxicity of the products made is missing.

The work is of interest as it explores the reuse of waste materials from a circular economy perspective. To obtain this result, however, it is important to determine the end of waste condition, satisfying the conditions of toxicity and physical-mechanical requirements of the raw materials to be replaced.

It is therefore essential to carry out the complete characterization of the materials before their reuse (MIBA, SSA) and that of the final product; in the case of this work, it is therefore necessary to better characterize the MIBA and SSA products and the finished product by subjecting it to release tests.

Thanks for the suggestion.  A final product TCLP should have make the study more complete.  However, since TCLP of the raw materials were done and found to be well below toxicity limits, we assume that less than 20% to 30%of MIBA or SSA mixing should not make the final products exceed the toxicity limit.  Please excuse us for omitting the TCLP for the final product.

In particular: 

üLine 124: what is the standard for industrial waste used in the table? Please add reference to the legislation. The standard concentration are expressed in mg/L?

We followed Taiwan EPA NIEA R201.15C for TCLP, which is correspondent to USEPA method 1311.  (added in Section 2.4).  The results were compared with Taiwan EPA “Standards for Defining Hazardous Indus-trial Waste” Table 4, and the standard concentration is expressed in mg/L in Table 4.

üLine 140: cm3, correct using superscript.

Font format has been corrected.

üFigure 4, 5, 6, 8, 9, 10: improve the resolution and the quality of the figure.

We have replaced the figures with 1000dpi resolution.

üParagraph 4, conclusions: correct oC into °C.

Celsius degree symbols were checked and corrected throughout.

Reviewer 2 Report

Den-Fong Lin et al. presented a novel study where two secondary waste materials: municipal incinerator bottom ash (MIBA) and sewage sludge ash (SSA) were used in addition to clay for ceramic manufacturing purposes.  The authors did an extensive characterization of the different combinations of MIBA, SSA and clay. They found that the best scenario for interior or exterior flooring applications would include 20% of SSA and 5 % of MIBA.

Initially, the authors need to double check the manuscript in terms of format (symbols, Figures #, etc.) in most of the sections. Some of them are listed below.

It is recommended that the authors address the following comments:

  1. In addition, the authors need to include detailed information about the equipment relevance of the methods, and the specification of the method by itself used in the material characterization (particle distribution, SEM images) and test results. For instance, a briefly description of the method and/or the respective reference is needed. This is very important especially for those colleagues who are interested in replicate your experiments or cite your research.
  2. An economic analysis should be added to justify the use of these secondary waste materials in addition to the enhancement of the different properties.
  3. A statistical analysis is requested to validate the differences among the different variables. In addition, the authors should provide information if the samples were run in duplicate or triplicate and add the respective deviation standard or error bar to the graphs.
  4. Although this a novel research, the discussion section should be improved through the addition of relevant references to justify or explain your results. There is only one reference after your section 3.2.

Additional comments

  1. Please check the symbol of Celsius degrees through all of your manuscript. The real symbol should be °C instead of ◦C or C. For instance: Line 70, Line 239, Line 329, Line 396, Line 405, etc.
  2. Line 32 -33 “… and early an early pioneering project and successfully delivered a successful ceramic processing of incinerator bottom ash”. The author should delete either successfully or successful.
  3. Line 65: Define the acronym LCD.
  4. Line 89: The authors mentioned “high temperature” please be more specific and provide the respective value or range.
  5. Line 132: the authors mentioned, “meeting the hazardous industrial waste standards”. Please clarify if the previous statement is for certain specific country or it is in general. Also add the respective information (for example as note) to the table 3, bottom line “standard”.
  6. Line 162: Correct Figure #
  7. Line 193: correct typo “.,”
  8. Line 217: add the respective concentration to the caption figure
  9. Line 267: it looks like the author added the wrong figures according from the “y axis”
  10. Line 287: provide an example about the code that the author used to identify the different scenarios in the main document.
  11. Line 297: Correct figure #
  12. Line 319: Change Al2O3 by Al2O3
  13. Line 327-333: The authors need to provide information about the meaning of Q0-Q4 to facilitate the understanding of the specific results of this section.
  14. Line 370: Please add a note about the meaning of the symbols used in the table
  15. Line 381: Please correct the “,” symbol in the water absorption column
  16. Line 429: reference 15 was not added to the list

Reviewer 3 Report

Page 2, Line 86: “Because MIBA was obtained from incinerated refuses at high melting temperature, lots of pores were produced on its surface”. This statement is not clear what is “refuses”. Are you trying to say incinerated residual or something else? The word “refuses” looks odd

2.2. SEM images for raw materials: Provide high resolution scalebar and magnification level. This information is in very small font size which is hard to read. I have provided a reference of one of the presentations of SEM images, you can follow that and create a scale bar with information that is clearly visible. If for some reason you are not able to recreate that, make sure you provide high resolution images in the revised submission. Refer scale bar of the SEM image in Figure 1 of this manuscript "Recycling biosolids as cement composites in raw, pyrolyzed and ashed forms: A waste utilisation approach to support circular economy." Journal of Building Engineering 38 (2021): 102199. Also increase the size of the SEM images in the manuscript if possible

Page 5, section 3.2 Shrinkage: You are relying on form of silica content (SiO2) for the reasoning behind the change in properties. It’s better to avoid speculation that you carry out XRD analysis on all the three raw materials i.e., MIBA, SSA and clay. XRD can show the relative proportions of amorphous silica content in the diffractograms. When you present the results make sure you don’t delete the background as the deletion of background sometimes remove the amorphous hump.

Page 6, 3.3. Weight loss on ignition: Addition of CHNS (carbon, hydrogen, nitrogen, sulfur) analysis will add valuable information about the organic content of the different materials i.e., MIBA and SSA.

Please provide XRD diffractograms as well in reference to Table 4

Conclusion 1: How does silica affect shrinkage, please provide any existing reference or any relationship that proves it. Regarding the increase in glass (amorphous) formation at high temperature there is no evidence provided. Please provide XRD diffractograms showing the increase in any amorphous hump or revisit this conclusion in light of the available information.

Conclusion 2: The addition of CHNS analysis can help in gauging the actual organic content present in both SSA and MIBA, that can help in getting the true reflection of both the materials  

Reviewer 4 Report

The study provides a mix of municipal incinerator bottom ash and sewage sludge ash for ceramic manufacturing.

The two abovementioned waste materials were mixed with clay by taking in consideration different bottom ash and sewage sludge ash replacement percentages.

The investigation of the suitability of the two waste materials mix to produce ceramic carried out in the present manuscript led to encouraging results. This work can be considered a good contribution to the current literature about the topic dealt.

A revision before publishing is recommended according with the following points.

  • - General comment

The authors should improve the literature review citing other research papers presented in international journals dealing with similar topic and extend the introduction with recent and relevant papers. See for example:

  • Ceramics from municipal waste incinerator bottom ash and wasted clay for sensible heat storage at high temperature. Waste and Biomass Valorization11(6), 3107-3120.

  • Pre-treatments of MSWI fly-ashes: a comprehensive review to determine optimal conditions for their reuse and/or environmentally sustainable disposal. Reviews in Environmental Science and Bio/Technology, 18(3), 453-471.

- Minor points

  • Regarding all tables and figures, please ensure to center all figures and tables together with all the related captions.

  • The presented abstract is enough informative about the carried out work but it should also state briefly the purpose of the research. An abstract is often presented separately from the article, so it must be able to stand alone, therefore a brief introduction of the topic is required.

  • In the sections 2 and 3 there are several lacks in the description of the experiments carried out. The authors should mention the machines used to perform the experiments and the standards used

  • The references are very few. The references section could be improved by considering recent and relevant researches to be mentioned for a more exhaustive introduction of the topic dealt as already suggested in the general comment.
  • Please check the conformity of the document with the guidelines of the journal (authors’ names or references for example)

Reviewer 5 Report

I suggest the publication of this work, after these revisions:

  • Lines 10-11: “a series of tests and analysis were conducted to investigate how the two materials affect the quality of the final product and to what extent.” is too general.
  • “MIBA is a light-weighted porous article with high water-absorbing characteristic [2]. 28 MIBA has a specific gravity between 1.5 and 2.3, and water absorbing ratio at 8% to 18%. 29 MIBA is also regarded as a mixture of calcium-rich compounds and other silicates en-30 riched in iron and sodium [3].” Avoid repetitions. Moreover, the Introduction lacks fluidity. Please revise it.
  • What are the novelty aspects of your research. Highlight these points in the end of the Introduction.
  • Section 3 should be renamed “Results and Discussions”
  • Figure 7 could be moved in Supplementary materials or Appendix.
  • More discussion about your results is necessary. Findings should not be only reported but also discussed and compared with previous literature results.
  • In conclusion, numbered list should be substituted by bullet list.
  • Please, add a nomenclature.
  • “Authors contribution”, “Data availability” statements should be added according to the Guide of Authors of the Journal.
  • I suggest you also more comparison with existing literature. It could add value to your research.

Round 2

Reviewer 1 Report

The answers are relevant to the comments.

In the conclusions, however, it would be appropriate to specify the only hypothesis that the final product does not exceed the TCLP. It is known in the literature that materials such as bottom ashes, subjected to heat treatments at high temperatures (1000 ° C) increase the leaching of some metals such as eg. chromium.

It is also suggested to include the CNS methods used in the bibliography.

Author Response

The Statement of the hypothesis is added in the conclusion (Line 517).  We will definitely conduct the TCLP for our final products to ensure their feasibility. Thanks for your advice.

CNS methods used are added in the bibliography [34] and [51].

Reviewer 2 Report

The authors addressed all of my comments. 

Author Response

We have checked the manuscript again and corrected all spelling errors. Thanks for your comment. 

Reviewer 3 Report

The authors' have addressed most of the comments. The manuscript can be accepted

Author Response

(The authors gave the same response as above.)

Reviewer 5 Report

In my opinion the paper can now be published.

Author Response

(The authors gave the same response as above.)
